# The role of water in host-guest interaction

Valerio Rizzi[1,2], Luigi Bonati [2,3], Narjes Ansari[1,2] & Michele Parrinello [1,2,4✉]

One of the main applications of atomistic computer simulations is the calculation of ligand binding free energies. The accuracy of these calculations depends on the force field quality and on the thoroughness of configuration sampling. Sampling is an obstacle in simulations due to the frequent appearance of kinetic bottlenecks in the free energy landscape. Very often this difficulty is circumvented by enhanced sampling techniques. Typically, these techniques depend on the introduction of appropriate collective variables that are meant to capture the system's degrees of freedom. In ligand binding, water has long been known to play a key role, but its complex behaviour has proven difficult to fully capture. In this paper we combine machine learning with physical intuition to build a non-local and highly efficient water-describing collective variable. We use it to study a set of host-guest systems from the SAMPL5 challenge. We obtain highly accurate binding free energies and good agreement with experiments. The role of water during the binding process is then analysed in some detail.

[1] Department of Chemistry and Applied Biosciences, ETH Zurich, 8092 Zurich, Switzerland. [2] Facoltà di Informatica, Istituto di Scienze Computazionali, Università della Svizzera Italiana, Via G. Buffi 13, 6900 Lugano, Switzerland. [3] Department of Physics, ETH Zurich, 8092 Zurich, Switzerland. [4] Italian Institute of Technology, Via Morego 30, 16163 Genova, Italy. ✉email: michele.parrinello@phys.chem.ethz.ch

Host–guest interactions regulate the working of proteins and have been intensively studied[1,2]. Atomistic simulations have been widely used[3–7] to calculate key parameters like ligand affinity and residence time, and to gain a microscopic understanding of how protein–ligand binding works. The accuracy of these simulations depends on two key aspects: the quality of the model used to describe the interatomic interactions and the thoroughness of the statistical sampling[8,9]. In this work, we will focus only on the latter and we will show that sampling can be much improved if the role of water in the binding–unbinding processes is duly taken into account.

Binding processes take place on a timescale that is unreachable with current computer resources; thus, the use of enhanced sampling methods is mandatory. We will frame our discussion in the context of Metadynamics (MetaD)[10–12] or, more precisely, of its most recent evolution, the on-the-fly probability-enhanced sampling method (OPES)[13]. OPES, like MetaD and many other methods[14–16], relies on the identification of suitable order parameters or collective variables (CVs). In these methods[14,17], the CV distribution is made to follow a preassigned law. This allows CV fluctuations to be amplified in a controlled way. For such methods to work in an accurate and efficient manner, the CVs must be able to describe the slow degrees of freedom of the system. Here we will identify one such powerful CV of general applicability aimed at describing the role of water in the ligand-binding process.

Water is expected to play an important role since, upon entering the binding site, the ligand has to shed its solvation shell in total or in part, while the water that originally was in the binding site has to rearrange and negotiate its way out of the binding cavity. Not surprisingly, much effort has been devoted to the role of water in ligand–host binding[18–24]. In the context of enhanced sampling, many attempts have been made at capturing the role of water in a CV, leading to an improvement in binding free energy estimations[5,25–28]. We show here that there is room for a further decisive step as none of these water-related CVs has been able to describe accurately the highly non-local changes in water structure that take place during binding, both in the vicinity of the ligand and in and around the binding pocket.

In order to succeed in our endeavour, we rely on a combination of physical considerations and modern machine learning (ML) techniques. In particular, we use a method that we have recently developed, which goes under the name of Deep Linear Discriminant Analysis (Deep-LDA)[29]. Deep-LDA builds efficient CVs from the equilibrium fluctuations of a large set of descriptors, expressing them as a neural network (NN). In this context, the choice of descriptors is essential and we appeal to our physical understanding to introduce one such set that is capable of characterising not only the ligand solvation shell but also the water structure inside and outside the binding cavity. After building such a CV, we use it in OPES for accelerating the sampling of binding–unbinding events.

We measure the performance of our approach on a set of test systems taken from the SAMPL5 competition[30–32] and study the interaction of six ligands with an octa-acid calixarene host (OAMe) (see Fig. 1). We choose this system because, despite its relative simplicity, it retains most of the key features of a biologically relevant protein–ligand system. Very recently, a closely related system has been used to investigate how water flows in and out of the system in the absence of a ligand[33]. Furthermore, the host's symmetry simplifies the analysis, and a comparison can be made to existing theoretical calculations[32]. The choice to perform simulations on a system with a standard set of simulation parameters allows our results to be compared to a range of different techniques, among which are the attach-pull-release method[34], alchemical protocols[35], and metadynamics[36].

## Results

**Collective variables from equilibrium fluctuations with Deep-LDA.** In this work, we are mainly interested in computing the free energy difference $\Delta G$ between the bound state (B) in which the ligand sits in the lowest free energy binding pose and the unbound state (U) where the ligand is solvated in water and free to diffuse. In order to obtain a CV able to capture water behaviour, we use the recently developed machine learning Deep-LDA method[29].

Deep-LDA is a non-linear evolution of the time-honoured Linear Discriminant Analysis (LDA) classification method[37]. In LDA, one takes two sets of data, in our case the configurations visited in short unbiased simulations in B and U, and defines a set of $N_d$ descriptors $\mathbf{d}$ that are able to distinguish between the two. The aim of LDA is to find the linear combination of descriptors $s = \mathbf{w}^T\mathbf{d}$ that best separates the two sets of data, $\mathbf{w}$ being an $N_d$-dimensional vector.

To this effect, one calculates for each set of data the vectors of the average descriptor values $\boldsymbol{\mu}_B$, $\boldsymbol{\mu}_U$, and their variance matrices $\mathbf{S}_B$, $\mathbf{S}_U$. With these quantities, one then computes the so-called Fisher's ratio:

$$\mathcal{J}(\mathbf{w}) = \frac{\mathbf{w}^T\mathbf{S}_b\mathbf{w}}{\mathbf{w}^T\mathbf{S}_w\mathbf{w}}. \qquad (1)$$

where one has defined the within–scatter matrix $\mathbf{S}_w = \mathbf{S}_B + \mathbf{S}_U$ and the between one $\mathbf{S}_b = (\boldsymbol{\mu}_B - \boldsymbol{\mu}_U)(\boldsymbol{\mu}_B - \boldsymbol{\mu}_U)^T$. The $\mathbf{w}$ that maximises this ratio is the direction that optimally discriminates the two states and gives the best-separated projection of the data in the one-dimensional $s$ space. The variable thus obtained has been shown to perform well as the CV in many cases, especially if one uses its Harmonic LDA variant[38,39].

In Deep-LDA, a similar paradigm applies with the key difference that LDA is performed on a non-linear transformation of the descriptors. The non-linearity is introduced by a neural network (NN) (see Fig. 2) whose input is the set of $N_d$ descriptors $\mathbf{d}$ and the outputs are the $N_h$ components of the last hidden layer $\mathbf{h}$. LDA is performed on the components of $\mathbf{h}$, so that, after determining the corresponding $\mathbf{S}_w$ and $\mathbf{S}_b$, the NN is optimised using $\mathcal{J}(\mathbf{w})$ as the loss function. At convergence, one determines the weights of the NN and the $N_h$-dimensional optimal vector $\mathbf{w}$ that produces the Deep-LDA projection:

$$s = \mathbf{w}^T\mathbf{h}. \qquad (2)$$

Deep-LDA is a powerful classifier that tends to compress the data into very sharp distributions which are unsuitable for enhanced sampling applications. To address this issue, we smooth the distributions by applying the following cubic transformation $s_w = s + s^3$, in the spirit of what was done in ref. [40]. The CV thus obtained will be used to describe the water behaviour in our simulations.

**Including water in the model.** The choice of the descriptors $\mathbf{d}$ is of paramount importance since it implies the physics that we want to describe. In our case, we are interested in capturing the role of water in the binding process. To this effect, we choose two sets of points around which we compute the water coordination number. One set is located on the ligand, while the other one is fixed along the host's axis $z$ at regular intervals (see Fig. 1 and the Supplementary Methods).

The first set of coordination numbers $\{L_i\}$ describes water solvation around the ligand and is similar in spirit to the ligand solvation variables that have been used in the past[5,28]. The second one $\{V_i\}$ is aimed instead at capturing the water arrangement inside and outside the binding pocket without any explicit reference to the ligand. It is essential that the descriptors capture all the water molecules that contribute to the host and the guest

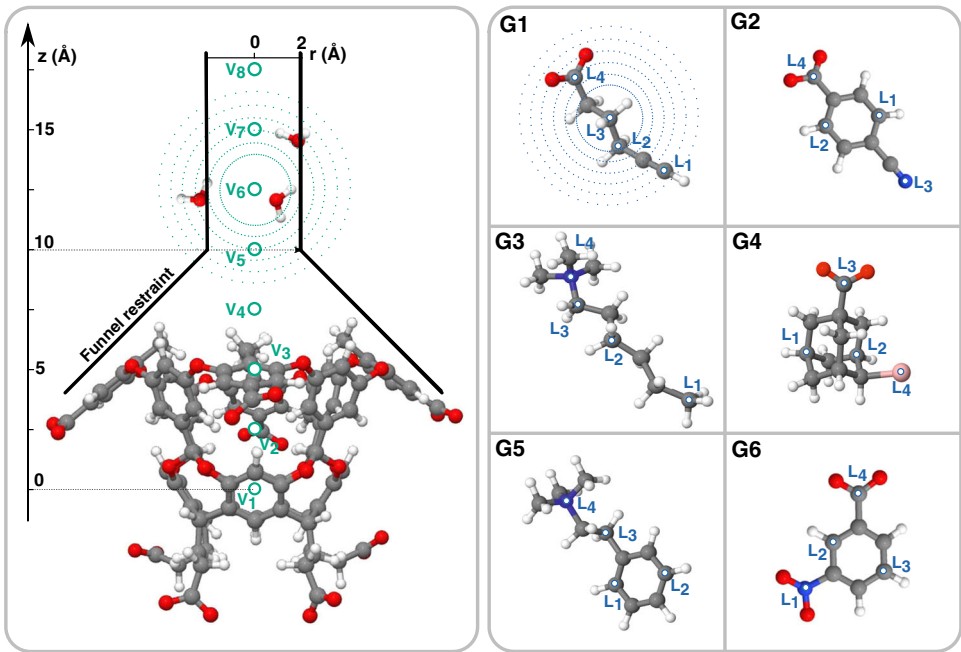

**Fig. 1 Sketch of the octa-acid host OAMe with the funnel restraint geometry and the guest molecules from the SAMPL5 challenge.** We indicate the position of the points where the descriptors are centred and hint at their spatial outreach by drawing surfaces at a constant radius around some of them.

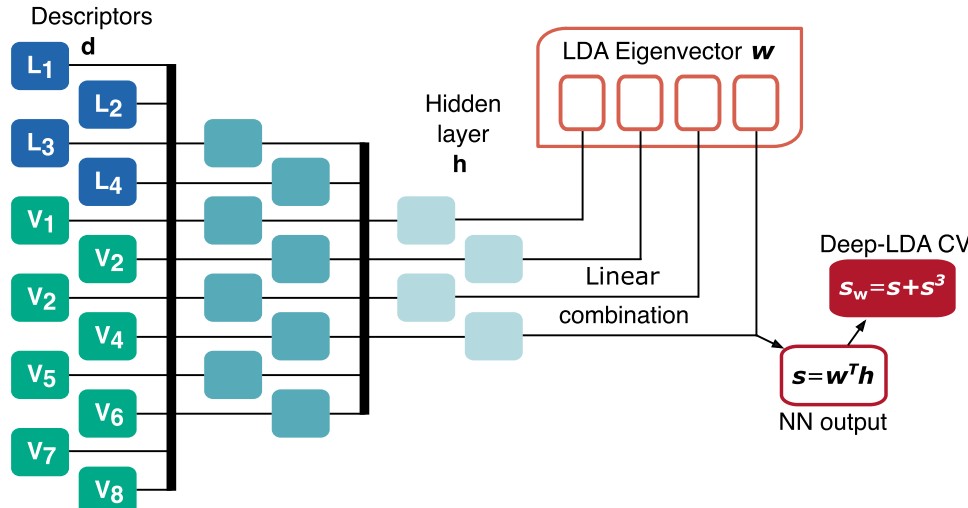

**Fig. 2 Schematics of the Deep-LDA architecture used in this work.** The descriptors **d** are fed to a NN that generates $s$ as a linear combination of the last NN hidden layer **h** and the LDA eigenvector **w**. The Deep-LDA CV is then $s_w = s + s^3$.

solvation. Missing some of them would create an incomplete picture of solvation, which in turn would lead to Deep-LDA classification errors and ineffective bias.

The set of descriptors $\{L_i, V_i\}$ gives information on the structure of water and its non-local changes on a small to medium length scale during the binding–unbinding process. Its effectiveness does not lie in the individual action of each descriptor but in its collective capability to capture the many-body concerted movements of the host, guest, and water molecules. The use of these descriptors is one of the elements of novelty in our approach and one of the keys to its success.

**Binding free energies from enhanced sampling simulations.** We perform OPES simulations to estimate the binding free energies of all the six ligands of Fig. 1. We use the Deep-LDA CV $s_w$ together with a second CV $s_z$, which is the projection of the

ligand centre of mass on the binding axis $z$. In the ligand-binding context, using the latter is a natural choice[5,36] as it has a clear physical interpretation and helps in distinguishing B from U. Furthermore, we employ a funnel-like restraint potential[4] to encourage the ligand to find its way back to the binding site once it is out in the solution. The entropic correction to the free energy due to the funnel restriction can be calculated analytically (see Eq. 4 in the Supplementary Methods) and is taken into account when computing the binding free energies $\Delta G$. We refer the interested reader to the Supplementary Methods for further details.

The combined use of these two CVs leads to an efficient sampling, which is reflected in a high number of binding–unbinding events per unit time (see for example Supplementary Fig. 18). We notice a clear improvement over a more standard set of CVs[36], namely $s_z$ itself, and the cosine of the angle $\theta$ between the binding axis $z$ and the ligand orientation (see Supplementary Fig. 17). The introduction of a water-

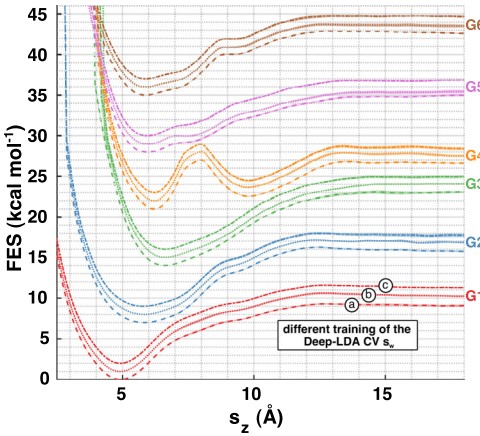

**Fig. 3 Free energy surfaces projected along the host–guest distance.** For each of the six ligands, we compute the free energy along the $s_z$ variable using a standard umbrella-sampling-like reweighting formula to recover the unbiased distribution[13]. The shaded areas indicate the errors, whose calculation is detailed in the Supplementary Methods. To ensure that the results do not depend on a specific realisation of the Deep-LDA CV, we repeat the training three times by using different initial weights of the NN. The resulting CVs are denoted as $s_w^a$, $s_w^b$, $s_w^c$, and the corresponding FES are indicated, respectively, by dashed, dotted, and dash-dotted lines. For clarity, curves related to the same ligand but with different CVs are shifted by 1 kcal mol$^{-1}$, while the shift between different ligand curves is 5 kcal mol$^{-1}$.

**Table 1 Binding free energies.**

| Ligand | Deep-LDA | Exp |
|---|---|---|
| G1 | −6.31 ± 0.06 | −5.24 |
| G2 | −6.19 ± 0.08 | −5.04 |
| G3 | −6.27 ± 0.07 | −5.94 |
| G4 | −2.51 ± 0.07 | −2.38 |
| G5 | −3.91 ± 0.09 | −3.90 |
| G6 | −4.97 ± 0.07 | −4.52 |

We show the mean binding free energy $\Delta G$ (kcal mol$^{-1}$) for every ligand and the corresponding experimental value. We calculate $\Delta G$ as a weighted block average over the simulations with all Deep-LDA CVs (see the Supplementary Methods for further details).

based CV in enhanced sampling simulations allows the system to reach a regime where it diffuses without hysteresis from one metastable state to another, yielding a high accuracy in estimating ensemble averages of physical quantities. This makes it possible to significantly reduce the error bars without having to increase the computational time relative to what is reported in the literature[34].

Performing enhanced sampling simulations allows retrieving the equilibrium distribution $P(s)$ of any collective variable $s$[14]. Here we focus on the free energy surface (FES), defined as $\mathrm{FES}(s) = -k_{\mathrm{B}}T \log P(s)$, where $k_{\mathrm{B}}$ is the Boltzmann constant and $T$ is the temperature of the system. In the context of ligand binding, it is customary to look at the FES as a function of the host–guest distance $s_z$. For each of the six ligands, we compute the FES and estimate the errors with a block average analysis. We report these results in Fig. 3, in which we also assess the robustness of the Deep-LDA CV by showing the results corresponding to three different rounds of Deep-LDA training.

We then report the binding free energies $\Delta G$ corrected for the presence of the funnel in Table 1. In Fig. 4 we compare them with experimental values and theoretical calculations performed on the same model but with different sampling techniques[34–36]. We assess the quality of our estimates through the metrics used in the SAMPL5 overview paper[32] and obtain a root-mean-squared error of 0.68 kcal mol$^{-1}$, a Pearson coefficient of determination of 0.93, a linear regression slope of 1.21, and a Kendall correlation coefficient of 0.87. With some exceptions, we are in line with the SAMPL5 results (see Fig. 4 and Supplementary Tables 1 and 2). However, the error bars are significantly reduced over the whole set of ligands investigated.

To test the generality of our procedure, we investigate the interaction of the six ligands with the OAH host also studied in the SAMPL5 challenge. The results are in agreement with those reported in refs. [34–36] and in Supplementary Figs. 31–57 and Tables 9–16 we provide a complete report. As a further check of our method and of the role of water, we also perform simulations of the host OAMe with the six ligands using the TIP4P/EW water model[41] instead of the TIP3P model[42]. While the binding/unbinding process is unchanged, we find that the binding free energies depend on the water model chosen. Modulo a shift of about 1.3 kcal mol$^{-1}$, the two sets of results correlate reasonably well with one another and with the experiments. For a

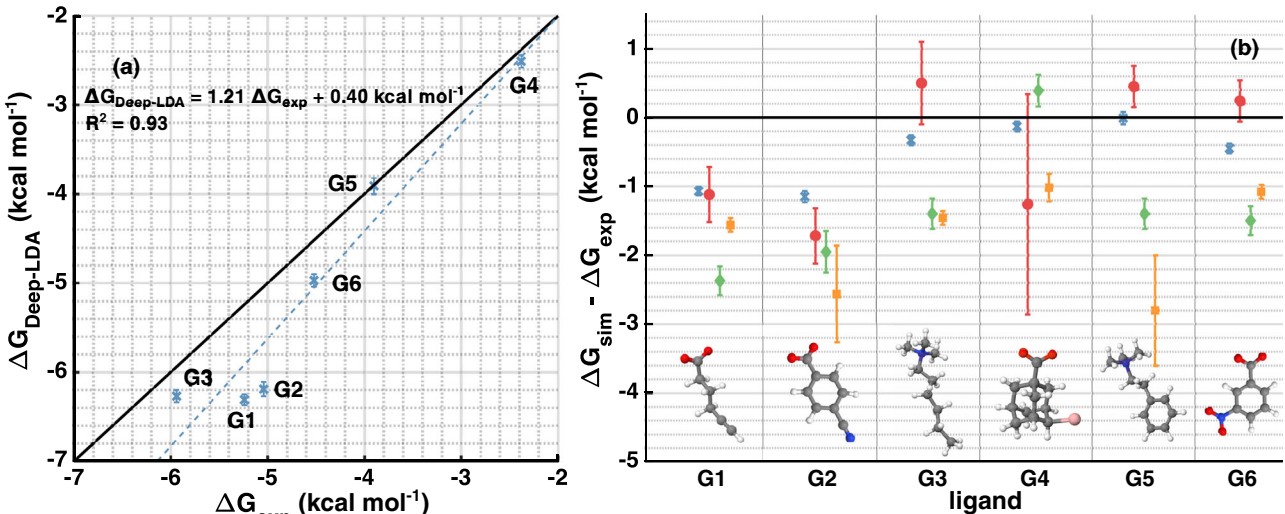

**Fig. 4 Comparison of the binding free energies with experiments and other calculations.** In **a**, we plot the value of $\Delta G$ obtained from the Deep-LDA simulations (in blue crosses) for every ligand versus the experimental values and show the corresponding linear fit. In **b**, we report their difference with the experimental values and compare them with other computational results performed using the same simulation setup. Results from ref. [36] are indicated with red circles, from ref. [34] in green diamonds, and from ref. [35] in yellow squares.

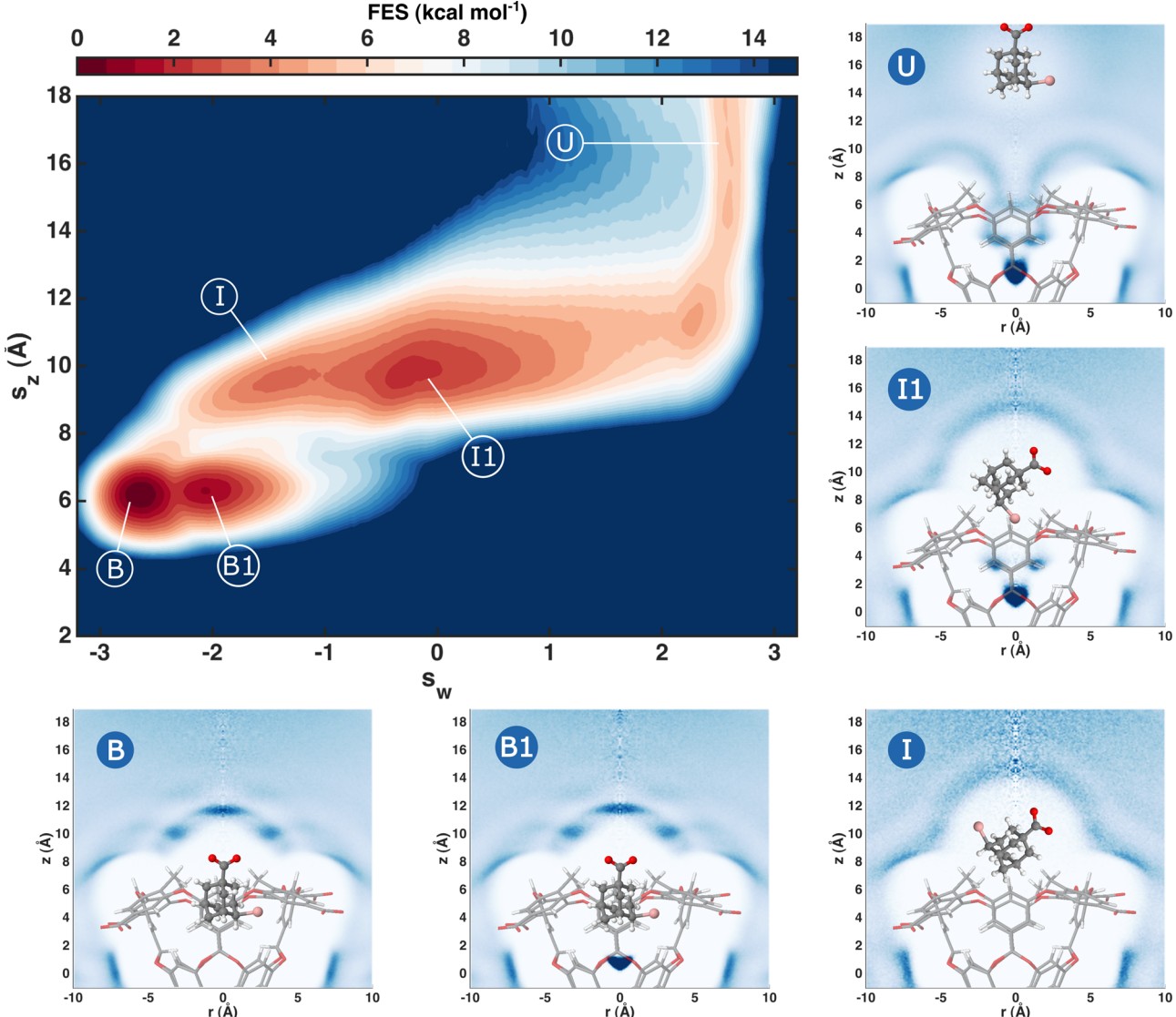

**Fig. 5 Binding FES of ligand G4 with a study of the water presence in the visited states.** We show the two-dimensional FES of the ligand G4 with respect to $s_z$ and Deep-LDA CV $s_w$. Different adjacent colours correspond to a free energy difference of $1\,k_BT \approx 0.6$ kcal mol$^{-1}$. We highlight some relevant states over which we perform plain molecular dynamics (MD) simulations to measure the presence of water. We show histograms of the water oxygen atoms' density in cylindrical coordinates $z$, $r$. Each histogram is normalised by the density value in its top-right corner and darker colours correspond to higher-water-density regions. The position of the ligand in these plots is illustrative.

quantitative assessment of these statements, see Supplementary Figs. 59–78 and Tables 19–26. The root of this change can be possibly attributed to a different solubility of the ligand in the two water models and to a different host–water interaction.

**The case of G4.** The use of the Deep-LDA CV $s_w$ allows us to obtain not only accurate binding free energies but also a detailed insight into water behaviour during the binding process. We illustrate here the case of G4, the guest that exhibits the most complex behaviour, and refer the interested reader to Supplementary Figs. 5–30 and Tables. 3–8 for a detailed analysis of all the other ligands.

In Fig. 5 we show the FES of G4 and the cylindrically averaged water density in the metastable states. We find that the system presents two binding poses B and B1. The lowest free energy binding pose B is the same as the one found in the experiments and contains no water. Our simulation discovered a second binding pose B1 that differs from B for the presence of a water

molecule at the centre of the cavity. This second pose is $\approx 2\,k_BT$ higher in free energy and thus it is occupied with a much lower probability.

When the ligand exits the pocket, before being fully solvated, it can pass through two intermediate short-lived states I and I1. In I, the cavity is dry and the ligand is free to rotate in front of the cavity entrance. In I1, the ligand sits again in front of the host entrance but its rotation favours configurations in which the ligand bromine atom points towards the cavity forming a linear arrangement where a water at the centre of the cavity is bridged by another water to the Br$^-$ anion (see Supplementary Fig. 21). We underline that neither B1 nor I and I1 were part of the Deep-LDA training.

The ability of the Deep-LDA CV $s_w$ to capture the non-local water structural changes is the main reason behind our capability to study the system's FES and its metastable states at this level of detail. For instance, the use of CVs that concentrate solely on the position of the ligand with respect to the binding site such as $s_z$ alone would clearly lead to an incomplete picture. In fact, B and

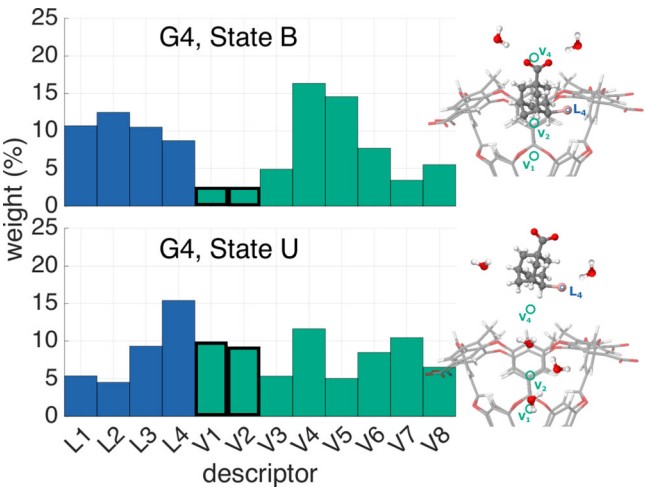

**Fig. 6 Descriptors' relative weights for guest G4.** Following the derivative-based ranking from ref. [29], we show the relative weight that each descriptor has in the Deep-LDA CV in the bound and unbound states of guest G4. We show the average weight over the three different Deep-LDA CVs that we trained. The $V_1$, $V_2$ descriptors, which measure the number of molecules inside the pocket, are outlined to mark the significant change in their contribution between the two states.

B1 (and similarly I and I1) cannot be distinguished properly by $s_z$ and, without the presence of a bias changing the cavity's solvation, the limiting timescale of the simulations would be the water movement in and out of the pocket. Furthermore, local CVs that only describe the average ligand solvation can only partially take into account these non-local effects.

**Analysis of the role of water**. We can gain a deeper insight into the role of water by investigating the dependence of the Deep-LDA CV on the $\{L_i, V_i\}$ descriptors. This can be done by analysing the descriptors' relevance in the action of $s_w$ and, for doing that, we use the derivative ranking method illustrated in ref. [29]. Here, we separate the role of the descriptors in the bound and unbound states and we report the results of ligand G4 in Fig. 6 (see Supplementary Fig. 22 for analysis over all the G4 metastable states).

In both states B and U, the weights are distributed over a wide range of descriptors, pointing to the fact that the Deep-LDA CV is able to capture the complex non-local action of water. However, different descriptors act in different ways in the two states. In the B state, the descriptors $V_4$, $V_5$, which are linked to the water molecules that reside in the proximity of the host's entrance, have more weight. This indicates that the fluctuations in this part of the water system need to be amplified for the ligand to exit.

In contrast, in the U state, the descriptors that gain more weight are $L_4$, which measures the solvation around the bromine atom of the ligand, and $V_1$, $V_2$, which control the quantity of water contained in the binding cavity. Fluctuations towards the dry state of the cavity need to occur for the ligand to bind. Such fluctuations can occur with a small but not negligible probability also in the holo state (see Supplementary Fig. 2). Even larger fluctuations have been observed experimentally in ref. [33] in a related system. We expect these fluctuations to be an important part of the reaction process in many host–guest systems.

The non-local action of the Deep-LDA CV is thus reflected in the relevance given to different water-based descriptors, depending on whether the system is in the bound or unbound state. When enhancing the sampling of this CV, this non-locality

determines a collective motion of water that encourages the occurrence of binding/unbinding events.

## Discussion

We have shown that, even in the relatively simple systems studied here, a complex and subtle reorganisation of water structure takes place and our strategy is able to capture it. Our calculations offer a powerful analysis tool and lead to accurate binding free energies.

Often, in the paper, we have underlined the efficiency of our method. However, this was not done in a spirit of competition with the SAMPL5 participants who, by the way, did not have the benefit of knowing the results beforehand. Our aim was instead to uncover and describe the role of water through the design and the application of an effective CV. In a scheme like MetaD, the efficiency of a CV is measured by its ability to capture the physics of the problem, hence our insistence on efficiency.

Having been able to reduce this much the sampling error on a commonly used model, we might even be tempted to claim that the discrepancies with respect to experiments can be blamed mainly on the inaccuracy of the force field. It would be interesting in this respect to investigate the force field limitations and how the inclusion of effects like polarisation could bring the results closer to experiments. The method is very robust and defines a protocol that can be naturally applied to larger and more complex systems. In fact, the sampling proficiency of our method will prove even more crucial in complex scenarios where a large number of water molecules can be trapped in multiple pocket locations.

**Reporting summary**. Further information on research design is available in the Nature Research Reporting Summary linked to this article.

## Data availability

The simulation inputs were taken from https://github.com/michellab/Sire-SAMPL5. We perform the simulations with GROMACS 2019.4[43] using the GAFF force field[44] with RESP charges[45] and the TIP3P water model[42]. For enhanced sampling, we use a custom version of the PLUMED plugin 2.5.4[46] where we include OPES[13] and the Pytorch library 1.4[47]. More details can be found in the Supplementary Methods. Simulations data are available on the Materials Cloud Archive at https://doi.org/10.24435/materialscloud:p3-1x.

## Code availability

All the inputs and instructions to reproduce the results presented in this manuscript are deposited in the PLUMED-NEST repository at plumID:20.025. A tutorial about the Deep-LDA training can be found at this link.

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

## Acknowledgements

We acknowledge the Swiss National Science Foundation Grant Nr. 200021_169429/1 and the European Union Grant Nr. ERC-2014-AdG-670227/VARMET for funding. This research was also supported by the NCCR MARVEL, funded by the Swiss National Science Foundation. The simulations were performed on the ETH Euler cluster. Many people helped us during the process of developing and writing this article. We give our sincere thanks to Sergio Pérez, Pablo Piaggi, Riccardo Capelli, Michele Invernizzi, Zoran Bjelobrk, Sandro Bottaro, Yue-Yu Zhang, Tarak Karmakar, Jayashrita Debnath, and Paolo Carloni. We also express our gratitude to the SAMPL challenges organisers for their precious initiative.

## Author contributions

V.R. performed the simulations. V.R., L.B., N.A., and M.P. discussed the results and reviewed the manuscript.

## Competing interests

The authors declare no competing interests.
