## [Peer Review File · Nature Communications]

REVIEWER COMMENTS

Reviewer #1 (Remarks to the Author):

In this manuscript, the authors apply a metadynamics enhanced sampling method to compute binding energies for a set of host-guest systems from the SAMPL5 challenge. The novel aspect of these simulations is the use of deep learning to determine a collective variable that incorporates the complex behavior of water and subsequent application of this collective variable in the metadynamics simulations. To determine the collective variable, a descriptor set involving the solvation coordination number of the unbound guest and host were fed as input into a neural network that was trained on host-guest configurations sampled from standard simulations. Results were then used to construct free energy surfaces, identify intermediate states, and compute binding energies. While the goal of computing highly accurate binding energies is an important one for molecular modeling, the manuscript focuses primarily on the machine learning algorithm and is therefore better suited for more specialized, methods-oriented journals such as the Journal of Chemical Physics or Journal of Chemical Theory and Computation.

To appeal to the broader audience of Nature Communications, the work would need to align with the title of the manuscript ("The role of water in host-guest interaction"), analyzing the role of water in more than one type of host-guest interaction (e.g. one or both of the two other SAMPL5 systems that are not covered in the manuscript) to yield more general fundamental insights. Furthermore, a much more in-depth analysis of the role of water would be required. For example, the authors could conduct a systematic analysis of the minimum amount of water information that is required in a collective variable for metadynamics simulations to reproduce experimental binding free energies along with a discussion of the physical intuition gained from the systematic analysis. In addition, they could test the use of different water models (e.g. 3-point, 4-point, and 5-point water models) to determine how the level of detail in the water model impacts the results regarding the role of water in the binding mechanism.

Below are more specific issues that should be addressed, regardless of the target journal.

- 1) To further highlight the importance of incorporating water behavior in the collective variable, it would be informative to include results from metadynamics simulations using a collective variable that leaves out descriptors involving water.
- 2) To demonstrate the robustness claimed about their method, the authors should discuss the convergence of their calculations, including convergence of the standard simulations that provided inputs to the deep learning algorithm, criteria for ensuring that the inputs for the Deep-LDA collective variable were reliable, convergence of the neural network, and convergence of the metadynamics simulations.
- 3) Figure 4: The significance of various metrics of the quality of the linear regression fit in Figure 4a such as the R² value, y-intercept, and χ^2 value should be discussed. In addition, the authors should comment on similarities and differences in the methodology used for their work and other studies reported in (ref. 29, 30, and 31), including reasons for why the uncertainties in their work are much lower than those from the other studies.
- 4) To appeal to a wider audience, I suggest that the authors conduct a thorough literature search on previous simulation studies of host-guest systems, citing more broadly than the metadynamics community in the Introduction, including other techniques used in the SAMPL challenges (e.g. ref. 29 and 30).

Reviewer #2 (Remarks to the Author):

This is an interesting paper with what appears to be an interesting new method. However, it is at present a bit too much of a sales pitch for the method, in my opinion.

Substantial points:

- I'd like to see more discussion of the limitations of the approach; in general, until I can understand when an approach WON'T work well or what its limitations are I tend to be skeptical. If anything the present piece serves as a bit of a sales pitch for the method. It's not dramatically overblown, but a bit so (especially in the conclusions), especially for a paper which doesn't compare to any other methods.
- The paper seems to have almost no recognition that accuracy is a function of not just the method but the force field employed (except perhaps line 162); for validation of a method, one would ideally compare with gold standard results for the force field, not with experiment. Certainly it is incorrect to attribute the "remarkable accuracy" (line 159) to the method employed, e.g. a poor method could actually yield better accuracy than a good one if sampling errors happen to offset force field errors.
- One of the conclusions is that the method is "very robust" but I see little data here that addresses robustness
- It is remarkable that the authors' work is enabled by the SAMPL organizers (and the Michel lab) making input files and other details available openly online, yet this paper does not make its own data available. The data should be made available in SI, not "upon request", to allow others to build on this work just as the authors themselves are doing.
- The SAMPL5 challenge itself should be cited, e.g. the overview paper — DOI 10.1007/s10822-016-9974-4 as well as other work the authors can compare to which uses the same FF, etc. It would likely be wise to also make contact with some of the many other methods which also participated in the challenge.
- I'd like to see some acknowledgment of the fact that the SAMPL5 challenge (and subsequent SAMPL challenges) was a blind challenge; obtaining accurate results retrospectively (as here) is fundamentally different from obtaining them prospectively (as when participating in a challenge). The authors should probably mention this to be fair to the SAMPL5 participants, and they may wish to consider participating in the SAMPL8 host-guest challenge, which is currently underway.
- The authors should perhaps consider validating their approach on the systems (and FFs) considered in the recent SAMPL6 "SAMPLing" challenge of Rizzi et al., as these systems allow isolation of sampling issues from force field issues in a way which is impossible for the present study.

Minor points:

- The authors are sometimes sloppy in using "binding energies" instead of "binding free energies" (eg abstract)
- Terms like "very efficient sampling" should be made quantitative rather than subjective, either by comparison to another message or by switching to some measure of efficiency. One person's "efficient" is another person's "inefficient" unless this is quantified somehow.
- In general try to avoid subjective, unquantifiable/unsubstantiated terms, e.g. "effortlessly" on line 102. The conclusions especially need policing for this.
- Be more clear what is meant by error, eg line 114 — error relative to experiment? Or relative to correct value for FF? The two are different in important ways.
- Relatedly, the force field is only mentioned very briefly in the methods section at the very end. It seems likely other SAMPL participants used the same force field, or a very similar one, in SAMPL5 so the authors should be able to make direct comparison to previously published results. This should be done to the extent possible; after all, this is one of the points of the SAMPL challenges. At present, it is peculiar that the authors don't set their results in the context of the broader challenge. What about comparing the results in Table 1 to what other methods obtained with the same, or nearly the same, force field?
- The authors should make sure to address the Gilson lab's work on water arrangements in host-guest systems like these and/or beta-cyclodextrins, as it's highly relevant.

Reviewer #3 (Remarks to the Author):

This is important and interesting work, employing machine learning to derive collective variables to describe host-guest binding and analysing the important issue of hydration changes via this approach in enhanced sampling simulations. There are good and thorough tests against a well-established test set, with encouraging results. The results and the methods will be of wide interest. The work is carefully described and appropriately referenced, with thorough analysis of the data and attention to significance and reproducibility. The work provides significant insight into hydration of binding, and the methods will find wide application. The manuscript is well written and is suitable for publication effectively as is. It could be worth commenting on limitations due to the forcefield and its functional form (e.g. lack of explicit polarisation changes, as recently investigated in other contexts by this group). It is very welcome that the relevant code is made available,

We would like to thank the reviewers for their helpful comments. We have carefully reviewed the manuscript according to reviewer's remarks. We are going to address each of the reviewers' points individually.

REVIEWER 1

To appeal to the broader audience of Nature Communications, the work would need to align with the title of the manuscript ("The role of water in host-guest interaction"), analysing the role of water in more than one type of host-guest interaction (e.g. one or both of the two other SAMPL5 systems that are not covered in the manuscript) to yield more general fundamental insights.

We have investigated the role of water in another SAMPL5 system, namely the OAH host interacting with the same six ligands. This host makes an interesting test case since OAH differs only slightly from the already studied OAMe. However, in some cases the binding free energies are rather different. All the results pertaining to this set are in Sec. S-3 in the SI. They are as well converged as those in the main text.

Furthermore, a much more in-depth analysis of the role of water would be required. For example, the authors could conduct a systematic analysis of the minimum amount of water information that is required in a collective variable for metadynamics simulations to reproduce experimental binding free energies along with a discussion of the physical intuition gained from the systematic analysis.

We have further investigated the role of water. An analysis of the role that the different descriptors play at different stages of the binding/unbinding process proved very illuminating. Our investigation shows how water fluctuations affect host-guest binding. The manuscript has been substantially changed to accommodate this discussion.

In addition, they could test the use of different water models (e.g. 3-point, 4-point, and 5-point water models) to determine how the level of detail in the water model impacts the results regarding the role of water in the binding mechanism.

We have tested our results against changing the water model from TIP3P to TIP4P and provided the results at page S31 of the SI. While the unbinding mechanism remains unaltered, the binding free energy does change, reflecting a significant role of water in binding.

1) To further highlight the importance of incorporating water behavior in the collective variable, it would be informative to include results from metadynamics simulations using a collective variable that leaves out descriptors involving water.

This information was already available in the SI of the original draft at page S21 for example. We make now a more explicit reference to this information in the new revision of the draft such that it is not any longer lost in the more than 50 pages of the SI.

2) To demonstrate the robustness claimed about their method, the authors should discuss the convergence of their calculations, including convergence of the standard simulations that provided inputs to the deep learning algorithm, criteria for ensuring that the inputs for the Deep-LDA collective variable were reliable, convergence of the neural network, and convergence of the metadynamics simulations.

There are two issues at play here. One is how precisely are the CVs determined. The other one is how well the metadynamics runs are converged. The determination of the CVs takes only a small fraction of the overall time and we have not bothered to optimise this part of the calculation, relying on the fact that a good CV leads to a speedy convergence. Instead, we went to great lengths to determine the statistical errors. We have clarified this issue at several places in the new manuscript.

3) Figure 4: The significance of various metrics of the quality of the linear regression fit in Figure 4a such as the R2 value, y-intercept, and Chi2 value should be discussed. In addition, the authors should comment on similarities and differences in the methodology used for their work and other studies reported in (ref. 29, 30, and 31), including reasons for why the uncertainties in their work are much lower than those from the other studies.

We apologise for the lack of clarity, but the line in Fig. 4 in the manuscript is not a fit but simply the bisectrix of the experimental versus theory plane. However, following the reviewer's recommendation, we have added the recommended fit to Fig. 4 in the revised manuscript. In the SI various metrics are used to assess the fit quality and the results are compared to those of the other SAMPL5 simulations. To facilitate this comparison, we used the bootstrapping procedure provided by the creators of the SAMPL5 challenge in Ref. 31.

4) To appeal to a wider audience, I suggest that the authors conduct a thorough literature search on previous simulation studies of host-guest systems, citing more broadly than the metadynamics community in the Introduction, including other techniques used in the SAMPL challenges (e.g. ref. 29 and 30).

We expanded the introduction by citing further reviews on the topic and explicitly mentioned a number of host-guest simulation methods.

REVIEWER 2

This is an interesting paper with what appears to be an interesting new method. However, it is at present a bit too much of a sales pitch for the method, in my opinion.

We are surprised at this remark. We thought that our wording was pretty tame considering the current level of salesmanship. However, we have further toned down the manuscript.

- I'd like to see more discussion of the limitations of the approach; in general, until I can understand when an approach WON'T work well or what its limitations are I tend to be skeptical. If anything the present piece serves as a bit of a sales pitch for the method. It's not dramatically overblown, but a bit so (especially in the conclusions), especially for a paper which doesn't compare to any other methods.

This again is a somewhat surprising observation since we have chosen to apply our method to the SAMPL5 system because among other reasons there were calculations one could compare to (namely the papers at Refs. 33-35). We also pointed out at page 4 some possible drawbacks of the method.

- The paper seems to have almost no recognition that accuracy is a function of not just the method but the force field employed (except perhaps line 162); for validation of a method, one would ideally compare with gold standard results for the force field, not with experiment. Certainly it is incorrect to attribute the "remarkable accuracy" (line 159) to the method employed, e.g. a poor method could actually yield better accuracy than a good one if sampling errors happen to offset force field errors.

We have clarified in a more expansive way that when comparing to the experiment there are two sources of error. One is the force field and the other one is the sampling accuracy. In our paper "remarkable accuracy" meant that the statistical errors could be brought down to a negligible value. The wording has now been changed in order to further tone down our claims. As far as we are aware of, in this area there is no gold standard. The best we could come up with was comparison with different methods as done in the SAMPL5 competition.

- One of the conclusions is that the method is "very robust" but I see little data here that addresses robustness

As a proof of robustness of the strategy, we added in the SI another set of simulations from the SAMPL5 challenge: the binding between host OAH and the same six guests. The results are of the same quality as the ones reported in the main text.

- It is remarkable that the authors' work is enabled by the SAMPL organizers (and the Michel lab) making input files and other details available openly online, yet this paper does not make its own data available. The data should be made available in SI, not "upon request", to allow others to build on this work just as the authors themselves are doing.

The material for replicating the simulations has been deposited in the public repository PLUMED-NEST plumID:20.025.

- The SAMPL5 challenge itself should be cited, e.g. the overview paper — DOI 10.1007/s10822-016-9974-4 as well as other work the authors can compare to which uses the same FF, etc. It would likely be wise to also make contact with some of the many other methods which also participated in the challenge.

The review paper mentioned above was already cited in the original manuscript [Ref. 26]. In order to give it more relevance, we analysed our results through the error metrics provided in Ref. 31 [pages 5-6 in the main text and SI]. We have also expanded the discussion of other methods at page 2. We also expressed explicitly our gratitude to the SAMPL5 organisers for their precious initiative.

- I'd like to see some acknowledgment of the fact that the SAMPL5 challenge (and subsequent SAMPL challenges) was a blind challenge; obtaining accurate results retrospectively (as here) is fundamentally different from obtaining them prospectively (as when participating in a challenge). The authors should probably mention this to be fair to the SAMPL5 participants, and they may wish to consider participating in the SAMPL8 host-guest challenge, which is currently underway.

In the revised conclusions, we have given due credit to the SAMPL5 blind competition participants.

- The authors should perhaps consider validating their approach on the systems (and FFs) considered in the recent SAMPL6 "SAMPLing" challenge of Rizzi et al., as these systems allow isolation of sampling issues from force field issues in a way which is impossible for the present study.

This is a good suggestion that we will consider in a future investigation.

- The authors are sometimes sloppy in using "binding energies" instead of "binding free energies" (eg abstract)

This has been corrected.

- Terms like "very efficient sampling" should be made quantitative rather than subjective, either by comparison to another message or by switching to some measure of efficiency. One person's "efficient" is another person's "inefficient" unless this is quantified somehow.

This is certainly a correct statement. However, in our case we have taken as state of the art the SAMPL5 results. Such a comparison shows that we can bring down the errors considerably investing a comparable amount of computer resources. In the spirit of toning down the self-promotion we have dropped the "very efficient" claim. We have clarified this point in the conclusions.

- In general try to avoid subjective, unquantifiable/unsubstantiated terms, e.g. "effortlessly" on line 102. The conclusions especially need policing for this.

The wording of this part has been changed.

- Be more clear what is meant by error, eg line 114 — error relative to experiment? Or relative to correct value for FF? The two are different in important ways.

We have clarified the part that the reviewer mentioned and added error metrics from the SAMPL5 challenge to measure the quality of our results.

- Relatedly, the force field is only mentioned very briefly in the methods section at the very end. It seems likely other SAMPL participants used the same force field, or a very similar one, in SAMPL5 so the authors should be able to make direct comparison to previously published results. This should be done to the extent possible; after all, this is one of the points of the SAMPL challenges. At present, it is peculiar that the authors don't set their results in the context of the broader challenge. What about comparing the results in Table 1 to what other methods obtained with the same, or nearly the same, force field?

We stress once more that we use the exact same force field setup as provided by the creators of the SAMPL5 challenge.

- The authors should make sure to address the Gilson lab's work on water arrangements in host-guest systems like these and/or beta-cyclodextrins, as it's highly relevant.

That is indeed relevant to what we do and we have added citations to recent work by Gilson's lab on improving sampling of trapped water in ligand binding

REVIEWER 3

This is important and interesting work, employing machine learning to derive collective variables to describe host-guest binding and analysing the important issue of hydration changes via this approach in enhanced sampling simulations. There are good and thorough tests against a well-established test set, with encouraging results. The results and the methods will be of wide interest. The work is carefully described and appropriately referenced, with thorough analysis of the data and attention to significance and reproducibility. The work provides significant insight into hydration of binding, and the methods will find wide application. The manuscript is well written and is suitable for publication effectively as is. It could be worth commenting on limitations due to the forcefield and its functional form (e.g. lack of explicit polarisation changes, as recently investigated in other contexts by this group). It is very welcome that the relevant code is made available,

We added a comment about the forcefield limitations in the conclusions.

REVIEWER COMMENTS

Reviewer #1 (Remarks to the Author):

The revised manuscript has been expanded to better suit the broad readership of the journal. However, there are a few remaining issues that should be addressed:

1) Table 1, S-2.3, and S-3.2 (Table 9): For the binding energy calculations involving the OAMe-G4 host-guest system, there is a considerable difference (6 kcal/mol) between TIP4P/EW data and TIP3P data. Large differences are also apparent between the energies calculated for ligands G1 and G4 with the OAH host. Please provide rationales for all of these differences.

2) S-2.3: The effect of using the TIP4P/EW water model for the simulations is inconclusive given that the water model was used for only one data point and this data point deviates considerably from the experimental value. I suggest that the authors perform TIP4P/EW calculations with the other ligands and whether their current data point is an outlier or a limitation of their computational strategy.

3) Line 23: "[...] the chosen CVs are amplified in a controlled way". The phrase "controlled way" is unclear and should be clarified.

4) Line 133: "While the binding/unbinding process is unchanged, we find that the binding free energy depends on the water model chosen (see SI). This is a further proof of the relevance of water." Since the "role of water" in the title, a more detailed discussion about the "relevance of water" is warranted, e.g. insights that can be gained from using a four-point vs. three-point water model.

Reviewer #2 (Remarks to the Author):

Overall, this work appears to be significantly improved, though I'm not entirely convinced by the authors' responses that they have done enough to tone down the overall "salesmanship" of this article (which they feel is apparently justified by the bad behavior of others -- "We thought our wording was pretty tame considering the current level of salesmanship."). I prefer articles to simply state the facts about performance, limitations, etc., without editorializing about whether this is "outstanding", "effortless", "superior" etc. Just give the statistics, compare the numbers to other methods, and let readers decide for themselves.

They also note that their explanation of their work as "very efficient sampling" is justified by comparison to state-of-the-art SAMPL5 results; SAMPL5 was conducted in 2016 and the community is now on SAMPL8, so it's not quite correct to describe these as state of the art.

Reviewer 1 suggested broadening the study to consider multiple host-guest systems, so they added one additional system (the sister system of that examined originally) which I don't think really addresses Reviewer 1's concern. (They could also have addressed the very SMALL number of tests in the SAMPL6 "SAMPLing" challenge for a much better assessment of their method, but they decided to defer this to another study.)

Overall, though, the authors may have done enough to address the earlier concerns.

We have carefully reviewed the manuscript according to reviewer's remarks. Our replies are reported below and we hope that the revised manuscript is now suitable for publication in Nature Communications.

We are going to address each of the reviewers' points individually.

REVIEWER 1

1) Table 1, S-2.3, and S-3.2 (Table 9): For the binding energy calculations involving the OAMe-G4 host-guest system, there is a considerable difference (6 kcal/mol) between TIP4P/EW data and TIP3P data.

In the first revised manuscript, the theoretical binding free energy for the OAMe-G4 host-guest system with the TIP3P water model that we report in Table 1 is 2.51 kcal/mol. The corresponding result with the TIP4P/EW model is reported in S-3.2 and is 3.87 kcal/mol. On the other hand, in S-3.2 (Table 9) we show the results pertaining to a different system: the OAH host. The difference in binding energy between water models that the referee mentions is not 6 kcal/mol but is about 1.5 kcal/mol. In the new manuscript, we perform an additional set of simulations in S-5 to better quantify the effect of the TIP4P/EW water model.

Large differences are also apparent between the energies calculated for ligands G1 and G4 with the OAH host. Please provide rationales for all of these differences.

This statement is not clear to us. Does the referee refer to the effect of using a different water model or to the relation between theory and experiment? We take that here the referee is interested in understanding the discrepancy between theory and experiment. However, if we take the metrics recommended by the referee in his first report ("The significance of various metrics of the quality of the linear regression fit in Figure 4a such as the R² value, y-intercept, and Chi² value should be discussed") we find that our data are aligned well along a linear regression fit.

2) S-2.3: The effect of using the TIP4P/EW water model for the simulations is inconclusive given that the water model was used for only one data point and this data point deviates considerably from the experimental value. I suggest that the authors perform TIP4P/EW calculations with the other ligands and whether their current data point is an outlier or a limitation of their computational strategy.

We have given careful consideration to the referee's remarks and performed extensive calculation on the OAMe system with all ligands with the TIP4P/EW water model. The binding energies shown in Fig. S-60 appear again to be well aligned and exhibit a systematic shift when compared to the ones resulting from the TIP3P water model simulations. The origin of this shift might be due to the different solvation energy and the effect that different water models can have on the host structure. Thus, the results reported earlier on OAMe-G4 were indeed representative of the TIP4P effects.

3) Line 23: “[..] the chosen CVs are amplified in a controlled way”. The phrase “controlled way” is unclear and should be clarified.

We have clarified this issue by changing the manuscript at page 1 by explaining the effect of enhancing the sampling along the chosen CVs.

4) Line 133: “While the binding/unbinding process is unchanged, we find that the binding free energy depends on the water model chosen (see SI). This is a further proof of the relevance of water.” Since the “role of water” in the title, a more detailed discussion about the “relevance of water” is warranted, e.g. insights that can be gained from using a four-point vs. three-point water model.

As discussed earlier, different models lead to different solvation energies and present slightly different hydrogen bonds strengths. We are afraid that dissecting all the role of these small variations have on a complex process such as this is beyond our capability and possibly those of most of other researchers.

REVIEWER 2

Overall, this work appears to be significantly improved, though I'm not entirely convinced by the authors' responses that they have done enough to tone down the overall "salesmanship" of this article (which they feel is apparently justified by the bad behavior of others -- "We thought our wording was pretty tame considering the current level of salesmanship."). I prefer articles to simply state the facts about performance, limitations, etc., without editorializing about whether this is "outstanding", "effortless", "superior" etc. Just give the statistics, compare the numbers to other methods, and let readers decide for themselves.

We would like to apologize to the referee for this inappropriate remark. In our minds, it was meant in a kind of tongue-in-cheek way. Of course he/she is right and in the new version of the manuscript we did remove all the trumpeting adjectives.

They also note that their explanation of their work as "very efficient sampling" is justified by comparison to state-of-the-art SAMPL5 results; SAMPL5 was conducted in 2016 and the community is now on SAMPL8, so it's not quite correct to describe these as state of the art.

In our effort to reduce the emphasis the adjective “very” was already taken out in the first revision and the “state of the art” statement only appeared in the answer to the referee’s question about efficiency. We agree that “state of the art” is a moving target. We chose to simulate the SAMPL5 set of systems because of the quantity of well established results that one can compare to and the availability of a prescribed simulation model. As our work is of methodological relevance, we think that its first application should be on a well known system. Now we are

indeed working on its application to more complex systems as we state in the conclusion.

Reviewer 1 suggested broadening the study to consider multiple host-guest systems, so they added one additional system (the sister system of that examined originally) which I don't think really addresses Reviewer 1's concern. (They could also have addressed the very SMALL number of tests in the SAMPL6 "SAMPLing" challenge for a much better assessment of their method, but they decided to defer this to another study.)

The paper now contains over 60 pages of Supplementary Information and the properties of 18 different systems have been simulated. This has amounted to tripling the number of systems studied relative to the original manuscript. We believe that there is enough material now for the reader to make his/her own judgement.